# Targeting the tsetse-trypanosome interplay using genetically engineered *Sodalis glossinidius*

**Linda De Vooght** [1,2]*, **Karin De Ridder**[1], **Shahid Hussain**[3], **Benoît Stijlemans**[3,4], **Patrick De Baetselier**[3], **Guy Caljon**[2]°, **Jan Van Den Abbeele**[1]°*

**1** Department of Biomedical Sciences, Trypanosoma Unit, Institute of Tropical Medicine Antwerp, Antwerp, Belgium, **2** Laboratory of Microbiology, Parasitology and Hygiene (LMPH), University of Antwerp, Wilrijk, Belgium, **3** Unit of Cellular and Molecular Immunology, Vrije Universiteit Brussel, Brussels, Belgium, **4** Myeloid Cell Immunology Lab, VIB Inflammation Research Center, Gent, Belgium

° These authors contributed equally to this work.
* linda.devooght@uantwerpen.be (LDV); jvdabbeele@itg.be (JVDA)

**Data Availability Statement:** All relevant data are within the manuscript and its Supporting Information files.

## Abstract

*Sodalis glossinidius*, a secondary bacterial symbiont of the tsetse fly, is currently considered as a potential delivery system for anti-trypanosomal components interfering with African trypanosome transmission (*i.e.* paratransgenesis). Nanobodies (Nbs) have been proposed as potential candidates to target the parasite during development in the tsetse fly. In this study, we have generated an immune Nb-library and developed a panning strategy to select Nbs against the *Trypanosoma brucei brucei* procyclic developmental stage present in the tsetse fly midgut. Selected Nbs were expressed, purified, assessed for binding and tested for their impact on the survival and growth of *in vitro* cultured procyclic *T. b. brucei* parasites. Next, we engineered *S. glossinidius* to express the selected Nbs and validated their ability to block *T. brucei* development in the tsetse fly midgut. Genetically engineered *S. glossinidius* expressing Nb_88 significantly compromised parasite development in the tsetse fly midgut both at the level of infection rate and parasite load. Interestingly, expression of Nb_19 by *S. glossinidius* resulted in a significantly enhanced midgut establishment. These data are the first to show *in situ* delivery by *S. glossinidius* of effector molecules that can target the trypanosome-tsetse fly crosstalk, interfering with parasite development in the fly. These proof-of-principle data represent a major step forward in the development of a control strategy based on paratransgenic tsetse flies. Finally, *S. glossinidius*-based Nb delivery can also be applied as a powerful laboratory tool to unravel the molecular determinants of the parasite-vector association.

## Author summary

Tsetse flies are the main vectors of African trypanosomiasis, a group of diseases caused by the protozoan *Trypanosoma* parasites which poses a severe burden on human/animal health and agricultural development in sub-Saharan Africa. The lack of prophylactic

**Funding:** The project work, LDV and GC were supported by the ERC-Starting Grant 'NANOSYM'(282312) awarded to JVDA. BS was supported by the Strategic Research Program (SRP3 and SRP47, VUB), Targeting inflammation linked to infectious diseases and cancer (Nanobodies for Health). The funders had no role in study design, data collection and analysis, decision to publish, or preparation of the manuscript.

**Competing interests:** The authors have declared that no competing interests exist.

drugs and limitations of the existing control programs urge the need for developing alternative strategies to complement the existing control programs. Paratransgenesis, the genetic manipulation of insect symbiotic microorganisms to block pathogen transmission, is a promising strategy for controlling vector-borne diseases. In this experimental study we successfully genetically modified the tsetse fly gut symbiont S. *glossinidius* to express trypanosome-interfering proteins (i.e. Nanobodies) thereby impairing trypanosome development in the fly. The application of the concept of using pathogen-targeting Nbs delivered by insect symbiotic bacteria could be extended to other vector-pathogen systems. Furthermore, our symbiont-based Nb delivery system can also be applied as a powerful laboratory tool to unravel the molecular determinants of the vector-pathogen association.

## Introduction

Vector-borne diseases are endemic in more than 100 countries and affect approximately 50% of the world's population. They are emerging and resurging and result in an unacceptably high burden of disease, especially in low- and middle-income countries, reflecting an inadequate implementation and/or impact from current control measures. Effective prevention strategies could reverse these trends, and vector control and pathogen-transmission blocking measurements through genetic engineering of the insect vectors are important components of such strategies. One of these devastating vector-borne parasitic diseases is African trypanosomiasis, caused by protozoan parasites of the genus *Trypanosoma*, including two human-pathogenic species of the *T. brucei* complex. African trypanosomes are biologically transmitted by the obligate blood feeding tsetse fly (*Glossina* sp.). The key to transmission is the intriguing and complex developmental cycle of the protozoan parasite in the alimentary tract of this haematophagous insect, where rapid transformation of the ingested bloodstream forms (BSF) into procyclic forms (PF) and subsequent establishment and proliferation in the tsetse fly midgut is a first prerequisite. The tsetse fly salivary gland is the final micro-environment where the *T. brucei* parasites adhere and undergo a complex re-programming cycle resulting in an end stage that is predestined to continue its life cycle in a new mammalian host [1].

Because of their viviparous lifestyle (*i.e.* embryonic and completed larval development occurs within the uterus of female flies), tsetse are recalcitrant to germline modification for the generation of transgenic fly lines refractory to trypanosome infection. The tsetse fly natural endosymbiont, *Sodalis glossinidius* resides in different tsetse tissues that are in close proximity to pathogenic trypanosomes, *i.e.* in the midgut [2] and can be cultured and genetically modified *in vitro* [3]. Therefore, a paratransgenic approach using *S. glossinidius* to deliver effector molecules that target the trypanosome-tsetse fly crosstalk offers possibilities to generate a trypanosome-resistant tsetse fly in which parasite development is blocked. Moreover, this approach can also be used to expand the currently limited experimental toolbox for functional research to unravel the molecular parasite-vector interactions that allow trypanosome development in the tsetse fly.

A crucial aspect for a successful symbiont-based paratransgenesis approach is the selection of effector molecules. Ideally, any effector molecule chosen to be secreted in the tsetse midgut by *S. glossinidius* should be highly effective, small, soluble, fast acting and resistant to midgut proteolytic digestion, without posing a significant fitness cost on the host in order not to be outcompeted by the wild type agent. Attention should also be paid to the potential development of parasitic resistance to the effector function. Nanobodies (Nbs), *i.e.* antigen-binding fragments derived from heavy chain-only antibodies (HCAbs) naturally occurring in

Camelidae and sharks [4,5], answer to almost all of these prerequisites. They are small in size (∼13–15 kDa), have a high stability and solubility, are expressed easily in microorganisms and show a superior tissue penetration compared to conventional antibodies. Candidate Nbs can also be mutagenized and selected for increased proteolytic stability [6]. Nbs targeting distinct epitopes of the variant-specific surface glycoprotein (VSG), abundantly present on the surface of BSF trypanosomes, have already been identified, including Nb_An46 which exerts a direct *in vitro* and *in vivo* trypanolytic activity by interfering with the complex endocytic machinery organized in the flagellar pocket of the parasite [7]. Recently, we demonstrated the proof-of-concept that *S. glossinidius* can be genetically engineered to express and release significant amounts of Nb_An46 in different tissues of the tsetse fly using a plasmid-based system [8]. Furthermore, we developed novel strategies for (i) establishing stable chromosomal expression in *S. glossinidius* [9] allowing strong and constitutive expression of anti-trypanosome Nbs in the absence of antibiotic selection, (ii) the sustainable colonization of the fly and its subsequent generation with genetically modified *S. glossinidius* through microinjection of the bacterium into third-instar larvae [10].

In the current study, we first tested recombinant *S. glossinidius* stably expressing a trypanolytic VSG-targeting nanobody i.e. Nb_An46 for its ability to block *T. brucei* development in the tsetse fly midgut. However, since BSF trypanosomes undergo a rapid transformation into insect-stage PF trypanosomes within several hours (2 to 4 h) following the fly's acquisition of an infective blood meal [11], we reasoned that recombinant *S. glossinidius* expressing Nbs targeting surface proteins of PF trypanosomes would have a much higher probability to hamper trypanosome development in the midgut. These Nbs would also benefit from the process of attrition that takes place approximately 3–5 days after infection. Indeed, in the early phase of a natural infection in the midgut, ingested trypanosomes undergo a severe population bottleneck (three orders of magnitude decrease), providing a unique window of opportunity for Nbs to target these early PF trypanosomes and interfere with their further development in the tsetse midgut. Since affinity is an important determinant of the anti-trypanosomal activity of Nbs [12], binding to an accessible, conserved epitope with sufficiently high affinity is necessary to achieve the required saturation. To meet these challenges, we generated a Nb library directed against procylic surface proteins by immunizing an alpaca with a mixture of purified EP-procyclin and a procyclic trypanosome membrane extract of which EP-procyclin represents a highly abundant surface protein of insect-stage PF trypanosomes known to cover the whole cell uniformly expressed by both early and late procyclic forms [13]. The membrane extract was included as an unbiased approach to target additional/new epitopes other than EP-procyclin. From this library we identified different Nbs that i) bind to the parasite surface and/or ii) compromise the *in vitro* growth of procyclic trypanosomes. These anti-procyclic Nbs were then selected for paratransgenic delivery by recombinant *S. glossinidius*. Finally, we assessed the effect on the fly's susceptibility to trypanosome infection. Delivery of different Nbs via *S. glossinidius* led to both a more resistant and susceptible phenotype, proving for the first time the ability of recombinant *S. glossinidius* to deliver effector molecules that are able to target the trypanosome-tsetse fly crosstalk, thereby interfering with parasite development.

## Results

### Paratransgenic delivery of a VSG-targeting Nb does not impair trypanosome development in the fly

Previously, we demonstrated the proof-of-concept that *S. glossinidius* can be genetically engineered to express and release functional anti-trypanosome Nb_An46 in different tissues of the tsetse fly, including the midgut using a plasmid-based expression system [8]. However, since

**Table 1. Microscopic examination of the proportion of male *G. m. morsitans* flies infected with *T. b. brucei*. Flies were harboring WT *S. glossinidius* or rec*Sodalis*::Nb_An46 that expresses the Nb_An46.** Differences in infection rates between WT *S. glossinidius* and rec*Sodalis*::Nb_An46 harboring flies were compared using Chi-square (two-sided) and considered significant if p-values were lower than 0.05.

| *Sodalis* strain | Infected/total # flies | | *p* value | |
| --- | --- | --- | --- | --- |
| | Midgut | Salivary gland | Midgut | Salivary gland |
| WT *Sodalis* | 12/46 | 10/43 | 0.49 | 0.53 |
| rec*Sodalis*::Nb_An46 | 13/46 | 10/46 | | |

such gene expression necessitates a sustained selection pressure (*e.g.* antibiotic supplementation) which is not feasible under *in vivo* conditions, we used Tn7-mediated transposition to generate a recombinant *S. glossinidius* strain chromosomally expressing Nb_An46 conjugated to a pelB secretion signal under the control of a constitutive lacZ promoter. Nb_An46 expression and secretion was confirmed by Western blot analysis (S1 Fig) and rec*Sodalis*::Nb_An46 showed growth kinetics comparable to a wild type (WT) *S. glossinidius* strain (S2 Fig). Since we have recently demonstrated that intralarval microinjection of rec*Sodalis* proves to be essential to achieve efficient colonization of tsetse fly tissues [10], rec*Sodalis*::Nb_An46 was introduced into third-instar larvae, collected immediately after larviposition using microinjection. Pupated larvae were allowed to hatch and a subset of teneral flies was subjected to qPCR to quantify the recombinant as well as the total *S. glossinidius* densities present in abdomen and thorax (S3 Fig). Another subset of teneral male flies were offered a parasitized blood meal. Twenty-eight days after the parasitized blood meal, flies were sacrificed for microscopic examination of the salivary gland and midgut tissue. Teneral flies, resulting from third-instar larvae injected with WT *S. glossinidius* were used as controls in the infection experiment. In tsetse flies harboring rec*Sodalis*::Nb_An46, no significant difference in midgut (*p* = 0.49) and salivary gland (*p* = 0.53) infections was observed compared to flies harboring WT *S. glossinidius* (Table 1).

## Generation and panning of an alpaca anti-procyclic *T. brucei brucei* surface immune nanobody library

Nbs that target the PF surface might have a higher probability to hamper trypanosome development at this first stage in the fly. Therefore, an anti-procyclic surface immune Nb-library was generated followed by a panning strategy to select for PF-targeting Nbs. For this, EP-procyclin was purified from *in vitro* cultured *T. b. brucei* PF parasites (S4A Fig). Additionally, two consecutive membrane extracts were obtained from *in vitro* cultured *T. b. brucei* PF parasites (S4B Fig). The second membrane extract (MemX2) was found to contain a lower abundance of soluble components as compared to the first membrane extraction (S4B Fig). Next, an alpaca was immunized with the purified EP-procyclin and the second membrane extract. Strong antibody responses, both in the conventional (IgG1) and heavy chain antibody (IgG2) isotypes, were recorded against the soluble extract, the two membrane extracts and purified EP-procyclin as coating antigens in an indirect ELISA (S5 Fig).

Starting from the peripheral blood lymphocytes of this immunized animal, a Nb library was constructed by a VHH-specific nested RT-PCR and by ligation of the obtained inserts into the phagemid vector pMECS. The generated library had a size of $1.2 \times 10^8$ individual transformants for which it was estimated that 96% harbored an insert of the correct size as determined by colony PCR. This library was panned to obtain Nbs against PF trypanosomes. Here, five different strategies were used including: panning against (i) solid-phase immobilized purified EP-procyclin (S6 Fig), (ii) solid-phase immobilized PF membrane extract (S7 Fig), (iii) solid-phase immobilized PF trypanosomes, (iv) live PF trypanosomes in suspension (S8 Fig) and (v) live, trypsinized PF trypanosomes in suspension (S9 Fig). The latter parasites were anticipated to

present a proteolytically trimmed antigenic repertoire as it might occur in the tsetse fly midgut lumen where various proteases are present for blood meal digestion. Only the strategy relying on solid phase immobilized trypanosomes (iii) was unsuccessful due to a low parasite coating efficiency and due to (partial) dissociation of parasites from the surface during the panning procedure as observed with microscopy. Consequently, no enrichment of antigen-specific phages was observed as determined by phage ELISA.

First, panning was conducted against purified EP-procyclin (S6 Fig) with a clear enrichment of EP-procyclin-specific phages in the consecutive panning rounds as determined by phage ELISA (S6B Fig). A total of 576 clones were evaluated in a periplasmic extract ELISA for binding onto EP-procyclin. The 30 ELISA-positive clones, mostly derived from the third panning round, yielded 7 different Nb sequences belonging to 3 different families based on the amino acid sequence within the complementarity determining region (CDR)-3 (S6D Fig). Clone Nb_EP_3.4 was retrieved 28 times. Flow cytometric evaluation of binding of the selected Nbs (n = 7) onto the surface of live parasites was found to be weak (S6C Fig). This suggests that the recognized epitopes are cryptic and poorly accessible for the Nbs. Parasite fixation with 2.5% p-formaldehyde did not significantly enhance Nb binding onto the parasite surface.

In another approach, the Nb library was panned against solid phase immobilized PF membrane extract 2 (S7 Fig). Phage ELISA revealed already efficient enrichment of antigen-specific phages in the second panning round (S7B Fig). A total of 288 clones were evaluated in a periplasmic extract ELISA for binding onto these membrane antigens. This yielded 56 non-ambiguous sequences including 18 different clones belonging to 13 different families (S7D Fig). Similar as for the anti-EP-procyclin Nbs, weak binding was observed onto live parasites using flow cytometry (S7C Fig). For several of these Nbs, parasite fixation with 2.5% p-formaldehyde significantly enhanced Nb binding to the parasite surface.

Panning of the Nb phage library against live procyclic *T. b. brucei* ProAnv parasites (S8 Fig) resulted in a marked enrichment, especially in round 2, of antigen-specific phages against the two PF membrane extracts and the purified EP-procyclin (S8B Fig). 74 positive clones were identified by flow cytometry using individual periplasmic extracts (S8C Fig). Binding appeared in general more pronounced than with Nb clones obtained in the two previous panning strategies. Binding for most of the positive clones was found to be insensitive to trypsinization of the parasite surface as well as to fixation with 2.5% p-formaldehyde. A total of 69 non-ambiguous sequences were retrieved resulting in the identification of 16 different clones belonging to 2 Nb families, with one family being over-represented by 15 out of the 16 clones that only display a limited number of substitutions in the Nb frameworks (S8D Fig).

Panning of the Nb phage library against trypsinized live PF parasites also resulted in a marked enrichment of antigen-specific phages against the two PF membrane extracts and the purified EP-procyclin (S9B Fig). 49 positive clones were identified by flow cytometry assessing the binding of individual Nbs onto the surface of trypsinized parasites (S9C Fig). This resulted in the identification of 9 different clones, belonging to 4 different families. The most abundant clone selected in the panning strategy against non-trypsinized parasites was also recovered in this procedure and an additional 3 Nb families were identified (S9D Fig).

Collectively, over the various panning strategies, 50 different clones belonging to 21 different families were identified. It appeared that the four approaches were highly complementary, yielding mostly different Nb clones and families (Fig 1). Based on the diversity of their amino acid sequences in complementarity determining region (CDR) 3, a selection of these Nbs were produced in *E. coli*, purified and tested in flow cytometry for binding onto fixed *T. b. brucei* PF using a two-step detection (Fig 2). Binding of Nb clones directed against the EP-procyclin was consistently at the limit of detection. Several Nbs identified by panning against live parasites and the membrane extract were binding at detectable levels to the parasite surface.

**A**

|  | Overall |
|---|---|
| **Posive clones** | 225/1248 (18%) |
| **Clones sequenced** | 197 |
| **Ambiguous sequence** | 23 |
| **Clone diversity** | 50/174 (29%) |
| **Family diversity** | 21/174 (12%) |

**B**

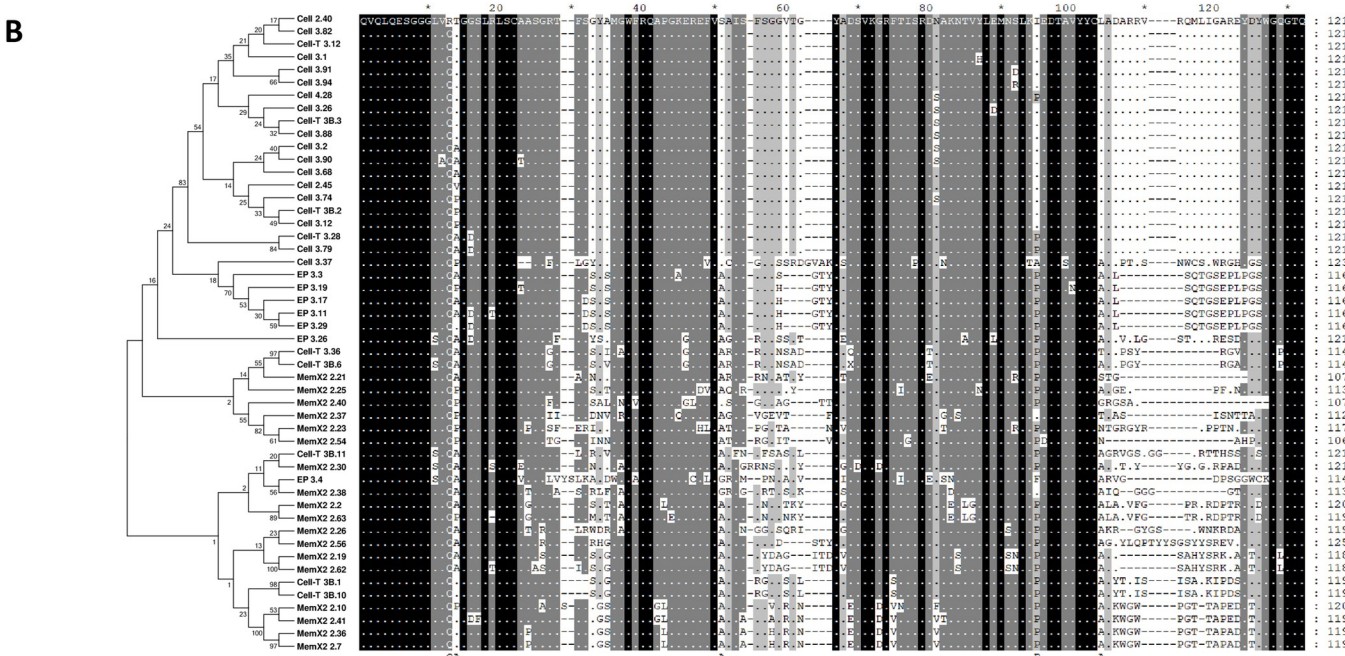

**Fig 1.** **(A)** Overview of the overall panning results obtained using 5 different approaches with numbers of positive clones and the resulting clone and family diversity. **(B)** Alignment of the different identified Nb clones with a corresponding maximum likelihood tree.

## Evaluation of the in vitro anti-trypanosomal activity of the selected anti-procyclic nanobodies

As a primary read-out, the activity of the different purified Nbs was evaluated in an *in vitro* toxicity assay against *T. b. brucei* AnTat1.1 PF parasites. Here, Nb_memX2_2.19 (Nb_19), Nb_memX2_2.36 (Nb_36) and Nb_memX2_2.63 (Nb_63) appeared to exert a consistent anti-trypanosomal activity (Fig 3A). This activity was mostly apparent from the third day of culture onwards, where parasite growth stagnated at relatively low densities or even decreased over time (Fig 3B). The three most potent anti-trypanosomal Nbs all resulted from the panning against the PF membrane extract.

## Engineering *Sodalis glossinidius* strains expressing anti-procyclic Nanobodies

The two most potent Nbs from the *in vitro* screen, *i.e.* Nb_19 and Nb_63, were selected for expression in *S. glossinidius*. Nb_cell_R3.88 (Nb_88), identified by panning against live *T. b. brucei* PF parasites, was also selected as it exerted strong binding to the parasite surface. The sequence encoding the respective Nbs were integrated into the *S. glossinidius* chromosome using Tn7-mediated transposition. Nb expression was confirmed by Western blot analysis

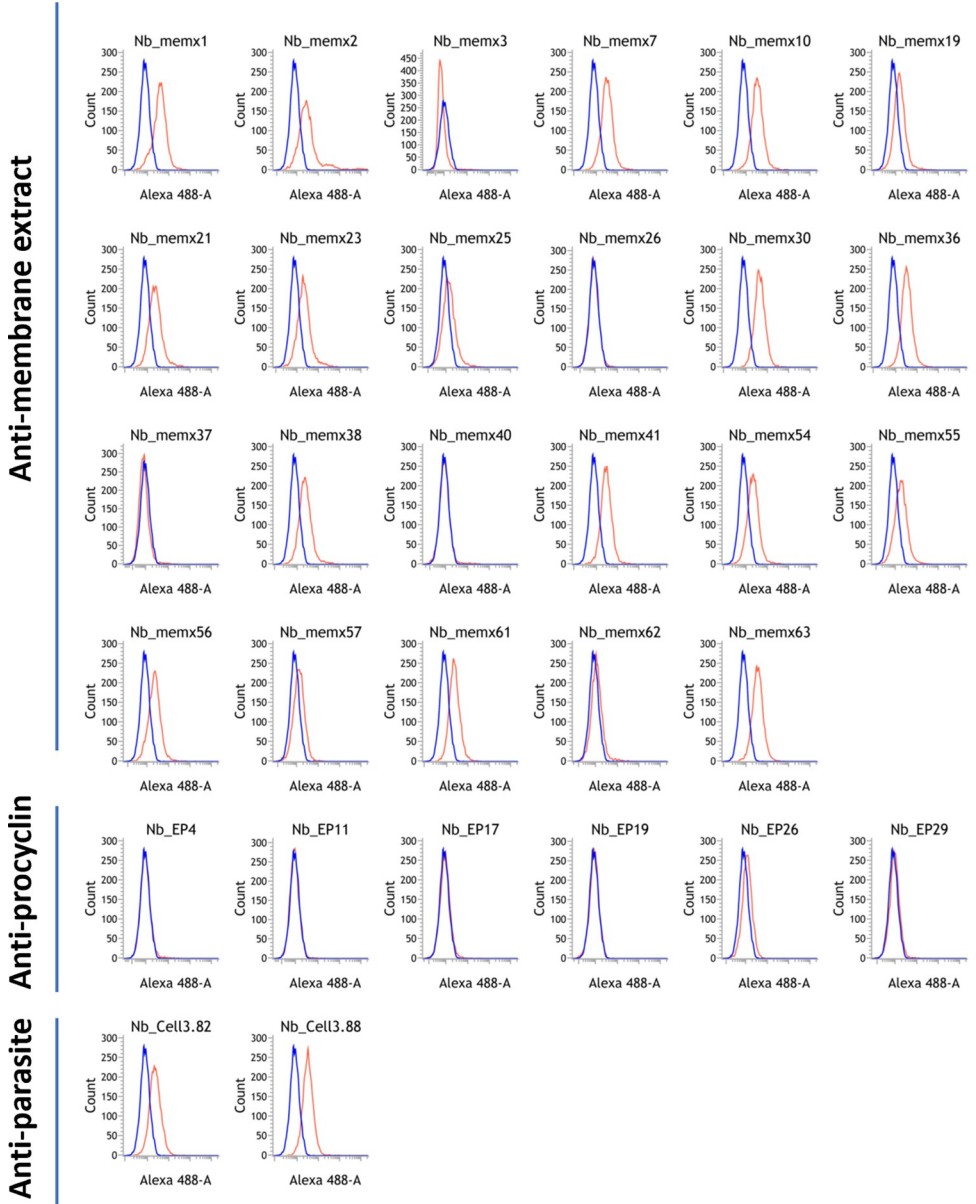

**Fig 2. Binding of individually purified anti-procyclic Nbs onto fixed procyclic *T. b. brucei*. The red histograms indicate staining with the respective anti-procyclic Nbs and detection using an Alexa Fluor 488 labeled anti-HA tag antibody.** A non-procyclic-specific Alexa488-labelled control nanobody (cAb-BCII-10) did not significantly bind to the trypanosome surface (blue histogram).

(S1 Fig) and recombinant *S. glossinidius* strains showed growth kinetics comparable to a WT *S. glossinidius* strain (S2 Fig). Since protein expression of Nb_63 could not be detected using Western blot analysis, rec*Sodalis*::Nb_63 was removed from further *in vivo* analysis.

## Recombinant *Sodalis glossinidius* strains affect trypanosome development in the fly

Prior to assessing the effect of the anti-procyclic Nbs released by rec*Sodalis*, we evaluated the effect of *per os* Nb delivery on tsetse fly susceptibility to trypanosome infection. For this,

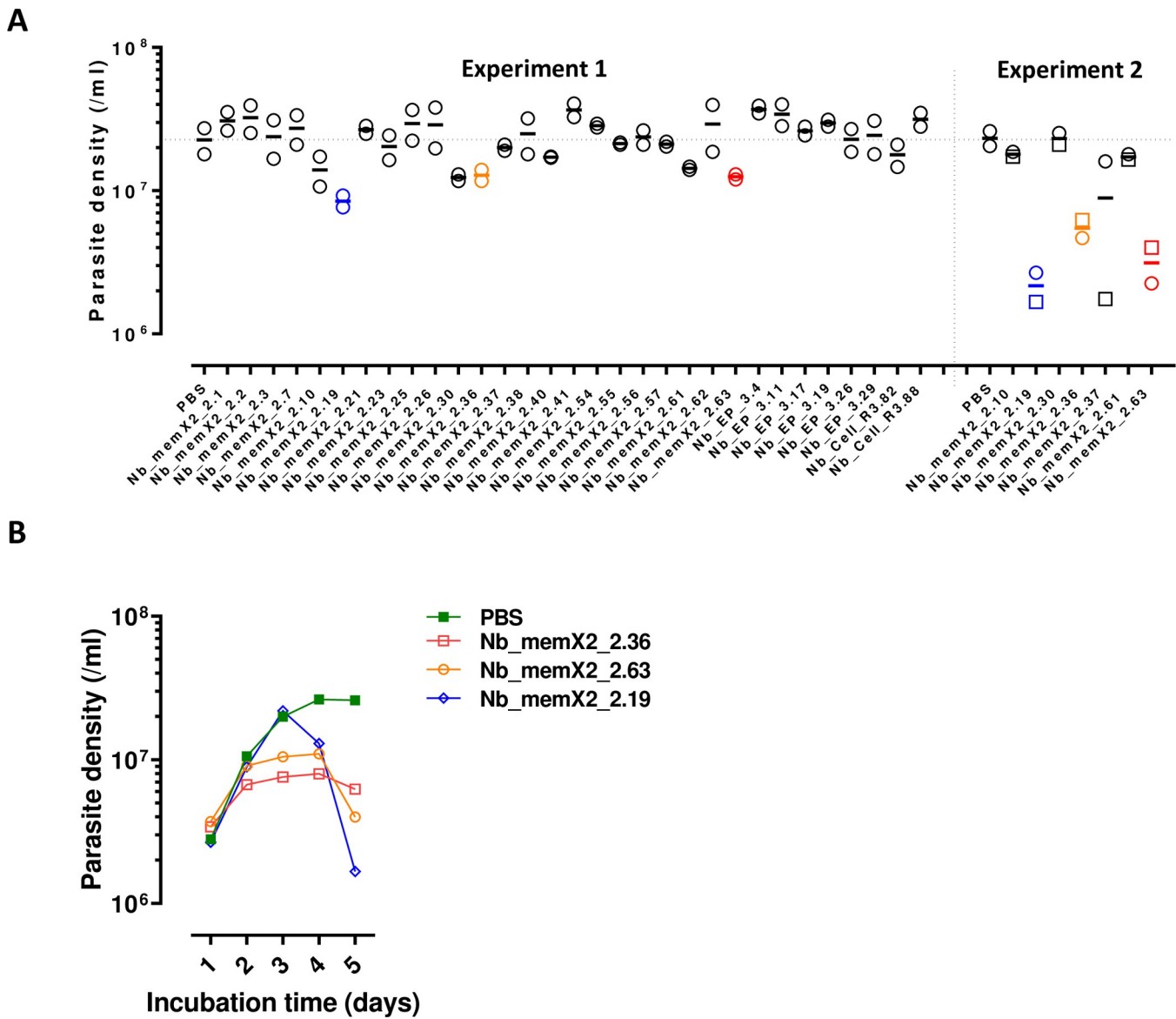

**Fig 3.** *In vitro* **toxicity assay with individually purified anti-procyclic Nbs.** (**A**) two separate experiments evaluating the toxicity after respectively 4 days (experiment 1; 10 µg/ml Nb, technical duplicates) and 5 days [experiment 2; 12.5 µg/ml (circles) and 25 µg/ml Nb (squares)]. Nbs with consistent anti-trypanosomal activity (Nb_memX2_2.19, Nb_memX2_2.36, Nb_memX2_2.63) are indicated in colour. (**B**) Parasite growth kinetics in cultures supplemented with 25 µg/ml of the respective anti-trypanosomal Nbs (Nb_memX2_2.19, Nb_memX2_2.36, Nb_memX2_2.63). Presented data are representative for at least two independent experiments.

purified Nb_19 and Nb_88 were administered by supplementation (at 25 µg/ml) of the primary blood meal (containing trypanosomes as well) and the following two feedings. A control Nb (Nb_23) binding to the parasites' coat (Fig 2) but without any apparent *in vitro* effect (Fig 3A) was also included. Midgut trypanosome establishment was examined by microscopy 3 days after the last Nb-supplemented blood meal (Table 2). Feeding with Nb_88 resulted in significantly lower trypanosome midgut infection rates while supplementation with Nb_19 resulted in a significantly higher proportion of midgut infected flies.

Next, recombinant *S. glossinidius* strains expressing Nb_88 and Nb_19 were evaluated for their ability to inhibit *T. b. brucei* development in the tsetse fly midgut. For this, third-instar

**Table 2. *In vivo* microscopic evaluation of the influence of a blood meal supplementation with 25 µg/ml Nb on the midgut trypanosome infection establishment.** The effect of each Nb was compared to a PBS control in independent experiments. Differences in infection rates were compared using Chi-square (two-sided) and considered significant if p-values were lower than 0.05.

| Feeding regime: blood meal supplementation | Infected/total # flies | *p* value |
|---|---|---|
| PBS control × 3 | 32/71 | |
| Nb_88 × 3 | **27/91** | **0.030*** |
| PBS control × 3 | 21/97 | |
| Nb_23 × 3 | 8/62 | 0.165 |
| PBS control × 3 | 20/110 | |
| Nb_19 × 3 | **34/115** | **0.046*** |

larvae were injected with the respective rec*Sodalis* strains. Teneral male flies resulting from the injected larvae (= paratransgenic flies harboring the respective rec*Sodalis* strains) were offered a parasitized blood meal and their midgut dissected for microscopic evaluation 8 days post infective bloodmeal (Table 3). In concordance with the results from the *in vivo* toxicity assay, *in situ* delivery of Nb_88 and Nb_19 via *S. glossinidius* resulted in significantly lower and higher midgut infections, respectively.

Finally, to assess more quantitatively the impact of the recombinant *S. glossinidius* strains on trypanosome development in the paratransgenic tsetse flies, the parasite loads were evaluated by qPCR 28 days after the infectious blood meal. In concordance with the above results, *S. glossinidius* expressing Nb_88 significantly reduced trypanosome density in the midgut ($p = 0.0017$), while *S. glossinidius* expressing Nb_19 resulted in higher trypanosome densities in the midgut ($p = 0.0095$) (Fig 4). In addition, the presence of rec*Sodalis* (*i.e.* the estimated amount of colony forming units (CFU)) at day 28 post infection in the midgut of these flies was also evaluated by qPCR (Fig 5). All flies were colonized with their respective rec*Sodalis* strain. Flies injected at their third instar larval stage with rec*Sodalis*::Nb_19, harbored the recombinant bacterium at persistent high densities (mean rec*Sodalis*::Nb_19 of $1,2 \times 10^6$ CFU (DNA equivalent)) while for rec*Sodalis*::Nb_88 the colonization in the different flies was more variable with densities ranging from $1 \times 10^3$ to $1 \times 10^6$ CFU (DNA equivalent) on the individual level (mean rec*Sodalis*::Nb_88 of $1,5 \times 10^5$ CFU (DNA equivalent)). Collectively, this experiment showed the ability of genetically modified *S. glossinidius* to deliver *in situ* effector molecules that are able of targeting the trypanosome-tsetse fly crosstalk thereby altering the infection outcome.

## Discussion

Genetically modified insect strains that are refractory to pathogen transmission are currently being explored as potential tools in the control of certain arthropod-borne diseases. During

**Table 3. *In vivo* microscopic evaluation of the influence of the presence of WT *S. glossinidius*, rec*Sodalis*::Nb_88 or rec*Sodalis*::Nb_19 on the midgut trypanosome infection establishment.** The effect of each rec*Sodalis* strain was compared to a WT *S. glossinidius* control in independent experiments. Differences in infection rates were compared using Chi-square (two-sided) and considered significant if p-values were lower than 0.05.

| *S. glossinidius* strain | Infected/total # flies | *p* value |
|---|---|---|
| WT *Sodalis* | 12/15 | 0.0457* |
| rec*Sodalis*::Nb_88 | 13/31 | |
| WT *Sodalis* | 17/50 | 0.0013*** |
| rec*Sodalis*::Nb_19 | 29/43 | |

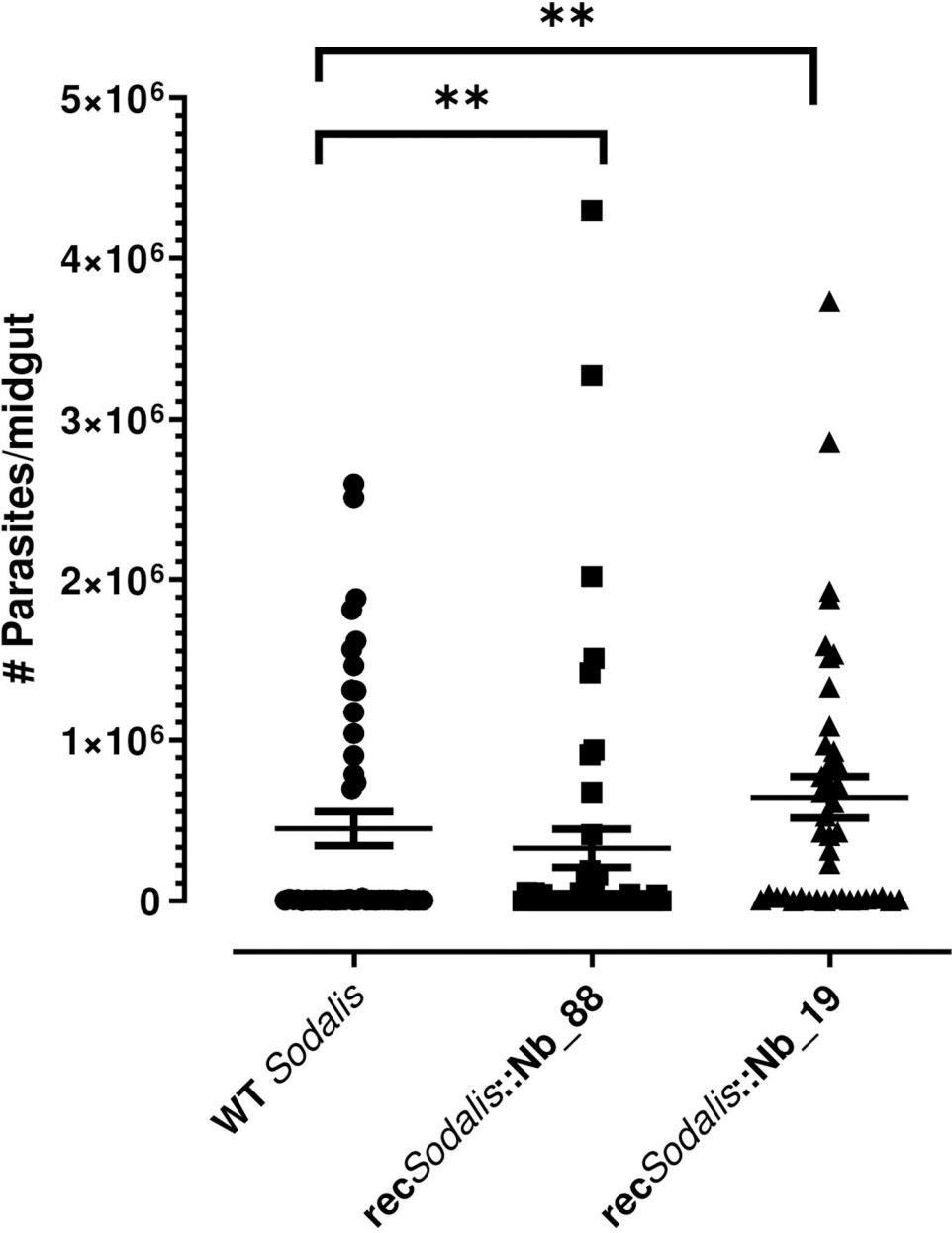

**Fig 4. Influence of recombinant *S. glossinidius* expressing Nb_88 and Nb_19 on midgut parasite densities determined by trypanosome-specific qPCR. p values were obtained by comparing the infection prevalence of the WT *S. glossinidius* (n = 48) harbouring group to the infection prevalence of either rec*Sodalis*::Nb_88 (n = 50) or rec*Sodalis*::Nb19 (n = 42) harbouring flies using a two-tailed Mann Whitney test.** Error bars represent mean with standard error of the mean (SEM) and are based on at least two independent experiments.

the last decade, considerable progress has been made in developing appropriate methodologies towards achieving this goal. Paratransgenesis is one such approach that aims to reduce vector competence by genetically modifying symbionts of disease vectors and has been demonstrated for *Rhodnius prolixus*, the triatomine vector of Chagas disease [14]. A paratransgenic approach in tsetse flies is also of high interest, especially since tsetse flies are not amenable to germ-line transformation techniques due to their viviparous lifestyle. At the heart of the development of a paratransgenic tsetse fly is the fact that the *S. glossinidius* symbiont and the trypanosome

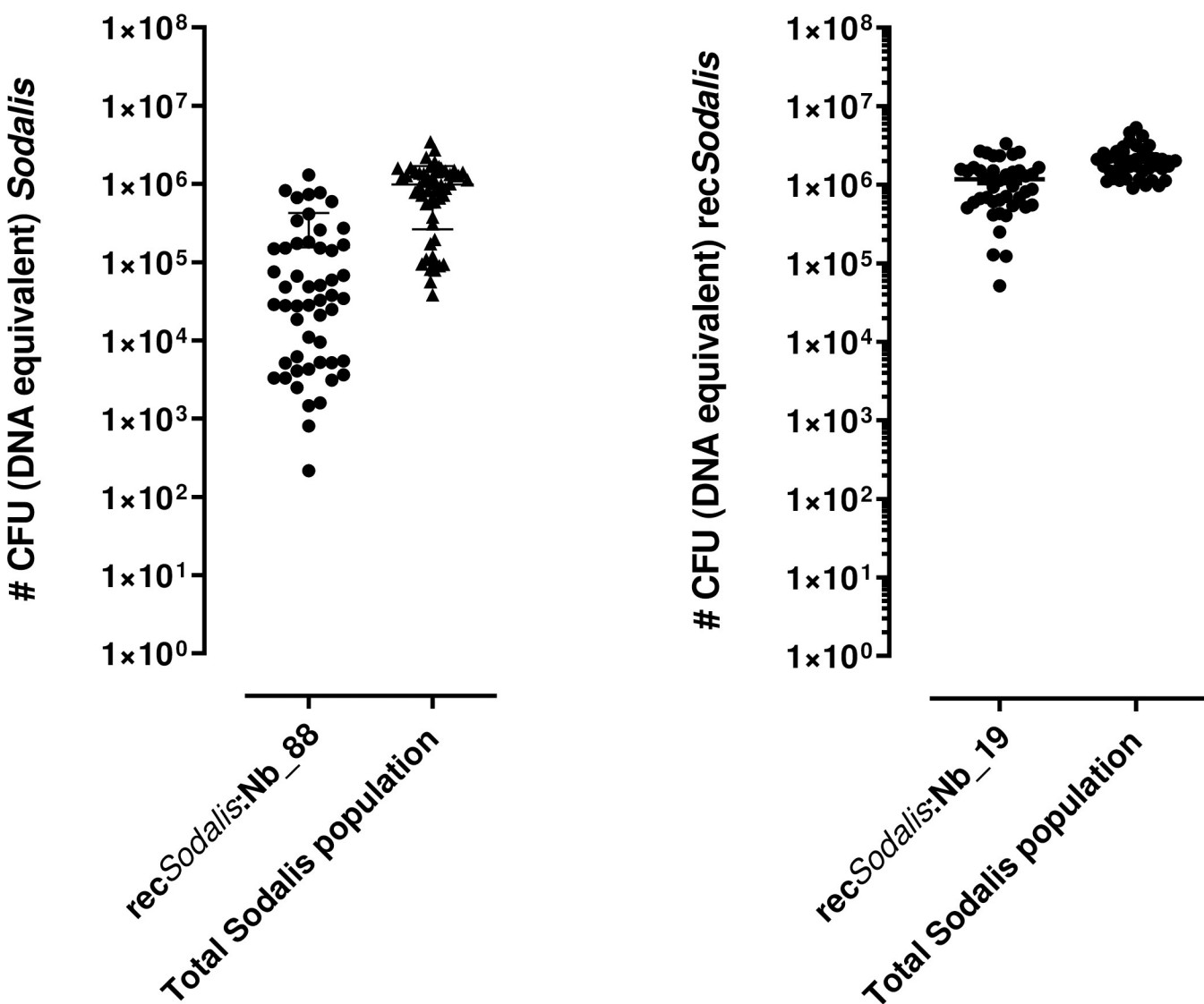

**Fig 5. Midgut characteristics of recSodalis colonized male flies.** Graphs depict the number of recSodalis CFU (DNA equivalent) versus the total number of Sodalis CFU (DNA equivalent) (WT + recombinant Sodalis) present in the midgut of 4 weeks-old male flies emerged from larvae injected with $5 \times 10^6$ recSodalis::Nb_88 (n = 50) (left graph) or recSodalis::Nb_19 (n = 42) (right graph). The number of WT and recSodalis CFUs was estimated using an already validated quantitative real-time qPCR protocol (8). The number of CFUs is represented in log scale on the y-axis. Error bars represent mean with standard deviation (SD) and are based on at least two independent experiments.

share the same compartment, *i.e.* the midgut where the most vulnerable stage of the parasite development occurs, making it a prime target for intervention. Here, ingested *T. brucei* parasites have to overcome a series of barriers to finally develop into the human infective metacyclic forms in the salivary glands. Because of numerous advantages such as small size, high stability and solubility, ease of expression and good tissue penetration, we investigated the use of Nbs as effector molecules to be expressed by *S. glossinidius* in the tsetse fly midgut. As a first candidate Nb_An46, a VSG-targeting Nb with known trypanolytic activity, was selected for expression by *S. glossinidius* in the tsetse fly midgut. Our results show that recSodalis: Nb_An46 had no impact on the tsetse fly's susceptibility to trypanosomes both at the midgut and salivary gland level. This result was somewhat expected as BSF trypanosomes differentiate

almost immediately into the PF after ingestion, which involves shedding of the VSG coat and its replacement by a new set of GPI-anchored proteins, *i.e.* procyclins [11] which is no longer recognized by Nb_An46. Therefore, we reasoned that Nbs targeting insect-stage procyclic antigens would have a longer window of opportunity for interaction with the parasite. For this, an anti-*T. b. brucei* surface PF Nb library was generated by immunizing an alpaca with purified EP-procyclin and a PF membrane extract. Using native antigens has the advantage of the selected binders recognizing the target proteins with their naturally occurring post-translational modifications. Putative Nb binders were enriched from the phage library using five different panning strategies, which led to the identification of 21 distinct classes of Nb binders based on their unique sequences within the complementarity determining region 3 (CDR-3).

In tsetse flies, newly emerged or teneral flies are considered the most likely to develop a mature, infective trypanosome infection with flies becoming almost refractory to infection after the very first blood meal (teneral phenomenon) [15]. A major limitation of previous studies involving paratransgenic tsetse flies is the fact that the symbiont was introduced into the adult stage necessitating the use of antioxidants such as glutathione or cysteine to artificially enhance midgut infections to levels at which changes in infection rates would be apparent [16]. Teneral flies represent a more relevant stage for evaluating the impact of rec*Sodalis* on vector competence which does not require artificial enhancement of infection levels, however it requires the recombinant symbiont's presence at that stage which is experimentally challenging. Recently, we developed a methodology for introducing rec*Sodalis* via microinjection of third-instar larvae resulting in stably colonized adult tsetse flies [10].

Due to the laborious nature of genetically modifying *S. glossinidius* and the technical challenging nature of its injection into the freshly born third-instar instar larval stage (which only lasts for +/- 30 minutes, after which the larvae pupate and cannot be injected anymore), we could only assess a limited selection of Nbs for their impact on trypanosome infection in the tsetse fly. Nb selection was based on their binding to PF trypanosomes and *in vitro* trypanotoxic potential. Paratransgenic tsetse harboring the rec*Sodalis*::Nb_88 strain were less susceptible to midgut infection and had significant fewer parasites in their midguts compared to control flies. Unexpectedly, we observed a higher trypanosome midgut infection prevalence in flies harboring rec*Sodalis*:Nb_19 compared to control flies infected with WT *S. glossinidius*. Additionally, a significantly higher midgut parasite load was observed in *Gmm*::rec*Sodalis*::Nb_19 flies compared to control flies. The outcomes of the recombinant *S. glossinidius* based delivery of Nb_88 and Nb_19 on fly midgut susceptibility were similar to observations in flies that were provisioned a blood meal containing the respective purified Nbs, however the *in vitro* toxicity assay did not prove to be indicative for the *in vivo* susceptibility phenotype. This could be explained by the fact that besides a direct effect, Nbs have the ability to exert indirect effects on the trypanosome by interfering with important intermolecular interactions between the parasite and the tsetse fly vector such as growth regulation, modulation of the parasite micro-environment, invasion of the peritrophic matrix and adhesion to the salivary gland epithelium [12] which would not be apparent in an *in vitro* toxicity assay. Additionally, for the toxicity assay non-trypsinized PF trypanosomes were used and since Nb_88 was selected by panning against live, trypsinized PF trypanosomes to mimic surface antigen trimming in the tsetse midgut [17] this might also explain the lack of *in vitro* activity.

Tsetse flies demonstrate a high resistance to trypanosome infection (*T. brucei* sp.) which is reflected in low midgut (<10%) and salivary gland (< 1) infection rate of wild caught flies. In the majority of flies, trypanosomes are eliminated from the gut within several hours after acquiring an infectious meal with only a few susceptible flies capable of facilitating disease transmission [11]. Survival in and colonization of the tsetse fly midgut and proventriculus are the first crucial bottlenecks for the trypanosome to overcome. Tsetse fly-derived factors

demonstrated to be involved during this part of parasite development are physical barriers including the peritrophic matrix [18,19], proteolytic enzymes, the immune deficiency (IMD) pathway regulated antimicrobial peptide attacin [20] and an immune-responsive midgut EP-protein [21]. Oxidative stress has also been suggested to play a role in fly infection as feeding of anti-oxidants significantly increases parasite establishment in the tsetse midgut but subsequently compromises further development into the final metacyclic forms in the salivary glands [22]. However, many questions regarding the molecular mechanisms underlying trypanosome development in tsetse fly largely remain unanswered due to the limited availability of experimental tools for functional research. Very recently, *S. glossinidius* expressing tandem antagomir-*275* repeats was used to investigate the role of tsetse miR275 in regulating key physiological processes such as blood digestion, PM integrity and gut environment homeostasis [16]. This indicated that a paratransgenic system could serve as a tool for studying functional mechanisms by knocking down gene function. Here we show that Nb_19 significantly enhanced trypanosome midgut infection rates. Although full characterization of the Nb_19-parasite interaction was beyond the scope of this study, addition of ALEXA-labelled Nb_19 to PF trypanosomes showed a clear staining of the flagellum (S10 Fig). Further characterization of the interaction between Nb_19 and PF trypanosomes could be assessed using pull-down assays and subsequent mass spectrometry-based protein identification which would shed further light on the underlying mechanism of the increased midgut susceptibility and potentially expose novel molecular determinants important for trypanosome development in the fly.

Collectively, we demonstrated the use of engineered *S. glossinidius* strains to interfere with trypanosome development in the tsetse fly. However, many challenges remain to achieve the long-term goal of controlling trypanosome transmission by paratransgenic modification of the fly. Since development of the parasite in the rec*Sodalis*-harboring tsetse fly was not completely blocked, it will be important to further select (a combination of) 'effector genes' capable of completely blocking trypanosome transmission. Importantly, we observed that the results of the Nb feeding experiments correlated to the outcome of *S. glossinidius* based delivery experiments which offers perspectives for a more straightforward initial selection of suitable effector molecules. The recent ability to genetically modify *S. glossinidius* using DNA transduction may facilitate the delivery of effector DNA to the bacteria in the host, thereby circumventing laborious culturing requirements [23]. Furthermore, the fact that *Sodalis* strains isolated from different tsetse species are similar enough to interspecifically colonize tsetse flies without any fitness cost [24] bypasses the need to genetically modify different tsetse fly-specific *Sodalis* strains, streamlining future paratransgenesis experiments. Although considerable efforts are needed to respond to these many challenges, the groundwork is laid for achieving trypanosome-resistant tsetse fly lines. Finally, we propose that the exploitation of Nbs that target the trypanosome-tsetse fly crosstalk represents a powerful new tool for investigating the molecular interactions that regulate this complex and dynamic relationship and by extension, can be applied to study pathogen-host interactions in the broader field of vector biology.

## Material & methods

### Ethics statement

Breeding and experimental work with tsetse flies was approved by the Scientific Institute Public-lic Health department Biosafety and Biotechnology (SBB 219.2007/1410). Animal ethics approval for the tsetse fly feeding on live animals was obtained from the Animal Ethical Committee of the Institute of Tropical Medicine Antwerp (Ethical clearance No. BM2012-6). The experiments, and maintenance and care of animals complied with the guidelines of the

European Convention for the Protection of Vertebrate Animals used for Experimental and other Scientific Purposes (CETS No. 123).

## Tsetse flies and trypanosome species

*Glossina morsitans morsitans* (Westwood) from the colony at the Institute of Tropical Medicine (ITM Antwerp, Belgium), originating from pupae collected in Kariba (Zimbabwe) and Handeni (Tanzania), were used in the experiments. Flies were fed on gamma-irradiated bovine blood using an artificial membrane system and maintained at 26° C with 65% relative humidity. The procyclic *T. b. brucei* strain ProAnv (originally derived from *T. b. brucei* AnTat 1.1 bloodstream forms) was used for the purification of the trypanosome antigens for alpaca immunization and for the Nb selection and evaluation in the *in vitro* trypanosome survival/ growth experiments. The pleiomorphic *T. b. brucei* AnTAR1 clone was used for the tsetse fly infection experiments to evaluate the *in vivo* activity of the selected Nbs.

## Bacterial strains, plasmids and culture conditions

The *S. glossinidius* strain used in this study was isolated from the hemolymph of surface-sterilized *G. m. morsitans* from the ITM colony. Cultures were maintained *in vitro* at 26°C in liquid Mitsuhashi andMaramorosch (MM) insect medium (PromoCell) supplemented with 20% (v/ v) heat-inactivated fetal bovine serum (Gibco). For cloning, *S. glossinidius* strains were cultivated on MM agar plates composed of MM medium (without FBS) solidified by autoclaving after the addition of 1% of bacto-agar. Blood agar plates were supplemented with 10% packed horse blood cells (IMP) and yeastolate (5 mg/ml) (Gibco). All solid cultures were maintained in micro-aerophilic conditions generated using the Campygen pack system (Oxoid) which provided 5% $O_2$, 10% $CO_2$, balanced with $N_2$) at 26°C. Where appropriate, antibiotics were added to the media at the following concentrations: 100 μg/ml of ampicillin and 50 μg/ml of kanamycin. *S. glossinidius* strains expressing respectively Nb_46, Nb_19, Nb_63 and Nb_88 were constructed by Tn7-mediated transposition as described in [9]. Briefly, plasmid pGRG25 features the *tnsABCD* genes expressed under control of the pBAD promoter and a multiple cloning site, flanked by the left and right ends of Tn7 and was used for transgene insertion into the chromosome of *S. glossinidius*. Expression is driven from the *E. coli Plac* promoter. The desired VHH coding sequences (Nb_An46, Nb_19, Nb_63 and Nb_88) were ligated in frame and downstream of the *Erwinia carotovora* pectate lyase B (pelB) signal sequence and upstream of the human influenza hemagglutinin (HA) tag, and 6×His tag. This *lac* promoter: *Nb* cassette was then cloned into the multiple cloning site (*Not*I and *Xho*I) of the pGRG25 plasmid. The pGRG25 plasmid harboring the desired Nbs was transformed into WT *S. glossinidius* cells using a heat shock method. Transposition was verified by PCR using primers that flank the *att*Tn7 site (GlmS_Fw: 5'–TATGAAGATTATTCCCCTGCCGCA-3'; PhoS_rev: 5'-CCATTTAGCGTAAACCGGCG-3') (S11 Fig) and by subsequent sequencing.

## Growth curve measurements

Logarithmically growing cultures were used to inoculate 25 ml of MM-medium to an optical density at λ = 600 nm ($OD_{600}$) of 0.005. Cultures of the different *S. glossinidius* strains were allowed to grow without shaking for 48 h and samples were taken every 24 h. Then, cultures were transferred to a shaking incubator and grown in alternating cycles (12 h) of static and shaking conditions. Two samples/day were taken during the exponential growth phase. Doubling times during this growth phase were calculated using the following equation: doubling time (in hours) = h × ln(2)/ln(c2/c1) where c1 is the initial concentration and c2 is the concentration when cultures reached maximum densities.

## Western blot analysis

Cultures were grown to the beginning of stationary phase (S. *glossinidius* $OD_{600}$ 0.5–0.6; *E. coli* $OD_{600}$ 1.5–2). Cells were pelleted from bacterial cultures by centrifugation (15 min, 10,000 × *g*) and the supernatant was clarified from residual bacterial cells by a second centrifugation step. Spent media from transformed *S. glossinidius* strains were concentrated using Micron 10 kDa filters. For SDS-PAGE, samples were heat denatured at 95˚C in the presence of SDS-PAGE loading buffer containing β-mercaptoethanol and analyzed on a 12% (w/v) poly-acrylamide gel. Proteins were transferred onto a nitrocellulose membrane (Whattman). After overnight blocking with 1% (w/v) bovine serum albumin, the membrane was incubated sequentially with a mouse anti-6×His tag IgG1 antibody (1:1000; Serotec) and a rabbit anti-mouse-IgG antibody (1:1000; Serotec) conjugated to horseradish peroxidase. In between these successive 2 h incubations, the membrane was washed with PBS 0.1% Tween 20. Thirty minutes after adding the substrate (TMB 1-Component Membrane Peroxidase Substrate, KPL) the reaction was stopped by washing the membrane with water.

## Preparation of the procyclic *T. brucei brucei* antigens

Antigens prepared for alpaca immunization included (i) purified EP-procyclin and (ii) a PF trypanosome membrane extract.

**Procyclin.** $5 \times 10^9$ *T. b. brucei* PF parasites were harvested from an SDM-79 (BioConcept) / 10% FCS *in vitro* cell culture at 27˚C. The trypanosome pellet was washed twice in 5 ml chloroform/methanol/water (1:2:0.8 v/v) with thorough mixing (resuspension) and sonication. Each time, centrifugation was carried out for 15 minutes at 4,000 × *g* (4˚C) to remove the lipid-containing supernatant (delipidation). The final pellet was dried under $N_2$ stream, followed by four subsequent extractions of procyclin in 1 ml 9% (v/v) 1-butanol in water. All supernates were collected after centrifugation for 15 minutes at 4,000 × *g* (4˚C). These supernates contained the procyclin and were dried in a DNA speedvac centrifuge (Savant). Pellets were redissolved twice in 1 ml 5% 1-propanol in 0.1 M ammonium acetate, each time with a collection of the supernate after centrifugation (10 minutes at 15,700 × *g* / 4˚C). EP-procyclin was purified by hydrophobic interactions chromatography (HIC) on a self-packed 1 × 13 cm octyl sepharose column (GE Healthcare) connected to an HPLC apparatus (Akta Explorer, GE Healthcare). Elution was carried out over a linear gradient from 5% 1-propanol in 0.1 M ammonium acetate to 60% 1-propanol and occurred at 26–27% 1-propanol. The presence of EP-procyclin was confirmed by assaying for the presence of sugar/glycoprotein/glycolipid content by orcinol reagent (Sigma) staining on a TLC-plate (Polygram SIL GUV, Macherey-Nagel) (S4 Fig). Positive fractions were pooled and checked on 12% SDS PAGE combined with protein revelation by silver staining (S4B Fig). The EP-procyclin identity was confirmed in Western blot and solid phase ELISA using a commercially available mouse anti-procyclin monoclonal IgG (Clone TBRP1/247, Cedarlane). EP-procyclin aliquots for immunization and panning were stored at -80˚C.

**Procyclic soluble and membrane extracts.** $5 \times 10^9$ *T. b. brucei* PF trypanosomes were harvested from the above-described *in vitro* cell culture. Parasite pellets were frozen at -80˚C and subsequently thawed and lysed by sonication in 20 ml 100 mM HEPES pH 6.9 containing a protease inhibitor mix (Complete, Roche) and 2 mM phospholipase inhibitor (N-α-p-tosyl-L-Lysine chloromethyl ketone, TLCK, Sigma). The soluble extract was the supernate obtained after a 30 min centrifugation at 100,000 × *g* at 4˚C. A first membrane extract which still contained soluble components was prepared by resuspending the pellet in 8 ml 2% octyl-β-D-glucopyranoside (Sigma) [in 100 mM HEPES pH 6.9 with TLCK and protease inhibitor mix] followed by a 60 min centrifugation at 100,000 × *g* at 4˚C. The supernatant was collected as the first membrane extract. A second membrane extract was obtained by resuspending the

remaining pellet in 2 ml 2% octyl-β-D-glucopyranoside solution and by collecting the supernatant after a 60 min centrifugation at 100,000 × $g$. Membrane extracts were dialysed twice against 250 ml phosphate buffered saline (PBS) containing 0.05% octyl-β-D-glucopyranoside. The extracts were subjected to SDS-PAGE electrophoresis and procyclin-specific immunoblot analysis using the available mouse anti-procyclin monoclonal IgG. Aliquots for immunization and panning were stored at -80˚C.

### *Alpaca immunization against procyclic* T. b. brucei *antigens*

The purified EP-procyclin and the second PF membrane extract (membrane extract 2) were immunized on opposite lateral sites of the alpaca body by weekly subcutaneous injections for six consecutive weeks of 500 μg membrane extract 2 and approximately 100 μg purified EP-procyclin in presence of the same volume of Gerbu adjuvant (GERBU Biotechnik). Peripheral blood was taken for the isolation of blood lymphocytes and to assess the antibody response against the trypanosomal antigens. Conventional and heavy chain antibodies were purified from the plasma by protein A and protein G Sepharose affinity chromatography. Conventional IgG1 antibodies were purified using protein G Sepharose, a wash step with 150 mM NaCl 0.58% acetic acid pH 3.5 to remove IgG3 and an elution using 100 mM glycine-HCl pH 2.7. Heavy chain IgG2 was purified by a negative selection on protein G Sepharose and a purification with protein A Sepharose followed by a selective elution at pH 3.5. As controls, the same IgG fractions were also purified through the same procedure from an alpaca immunized against a non-related antigen. Reactivity of the purified antibodies was evaluated in an indirect ELISA developed using an in-house rabbit anti-camel polyclonal IgG and a peroxidase-conjugated anti-rabbit IgG (Sigma).

### Generation of an alpaca anti-procyclic *T. brucei brucei* surface immune nanobody library

From the alpaca immunized against the above-described *T. b. brucei* procyclic membrane components, lymphocytes were isolated from peripheral blood ½ diluted in RPMI1640 using Lymphoprep (Nycomed). After washing the lymphocytes once with PBS, RNA was extracted using Trizol reagent and cDNA was prepared from 32.2 μg total RNA for the construction of the anti-procyclic surface immune Nb library. VHH gene fragments ranging from framework region 1 to framework region 4 were amplified by nested PCR. The first PCR round (32 cycles 1 min at 94˚C, 1 min at 55˚C and 1 min at 72˚C) was performed with primers *callI* (5'-GTC CTGGCTGCTCTTCTACAAGG-3') and *callII* (5'-GGTACGTGCTGTTGAACTGTTCC-3'). PCR products were separated on a 2% agarose gel and the 700 bp band corresponding to the amplified VHHs fragments was excised and purified using the QIAquick gel extraction protocol (Qiagen). Using 10–50 ng of the purified first PCR product as template, the 2nd round PCR was performed using the same PCR conditions as mentioned above with primers *A6E* (5'-GATGTGCAGCTGCAGGAGTCTGGRGGAGG-3') and *pmcf* (5'-CTAGTGCGGCCG CTGAGGAGACGGTGACCTGGGT-3'). After purification using the QIAquick gel extraction protocol (Qiagen), nested PCR products and the pMECS phagmid [25], derived from pHEN4 [26], were digested with *Pst*I (200 U, Roche) and *Not*I (200 U, Roche). After purification, both the insert and pMECS were subjected to another round of overnight digestion at 37˚C using 80 U of *Pst*I and *Not*I. To reduce the risk of self-ligation during the library construction, the pMECS phagmid was subjected to an additional digestion by 80 U *Xba*I for 1 h at 37˚C followed by a column purification (Qiagen). Insert and phagmid were purified using the QIAquick purification kit (Qiagen) and unidirectional ligation was with 52.5 U T4 ligase (Invitrogen) for a total of 17.6 μg of pMECS vector and 18.6 μg of insert in a 20 μl reaction.

The ligation reaction (5 µl) was subsequently transformed with a 1.8 kV pulse into electrocompetent *E. coli* TG1 cells (Immunosource) using a BioRad GenePulser. Transformed cells were collected in recovery medium (Immunosource) and incubated for 1 h at 37˚C under non-shaking conditions, before plating on LB agar plates supplemented with 2% glucose and 100 µg/ml ampicillin. Dilutions were plated to determine the library size and to determine insert frequencies based on a colony PCR (32 cycles of 1 min at 94˚C, 1 min at 55˚C and 1 min at 72˚C) using primers *MP57* (5'-TTATGCTTCCGGCTCGTATG-3') and *GIII* (5'-CCACAGACAGCCCTCATAG-3'). The library was collected in LB-medium supplemented with 20% glycerol, aliquoted and stored at −80˚C.

## Selection of anti-procyclic *T. brucei brucei* nanobodies through panning

The anti-procyclic *T. b. brucei* surface immune library was expressed on phages after super-infection with M13K07 helper phages (Invitrogen). Libraries were enriched by several consecutive rounds of *in vitro* selection by five different strategies (explained further below) and using the same phage-enrichment and binding-assay methodologies. Phages were eluted under alkaline conditions with triethylamine (pH 11.0), followed by an immediate pH neutralization by the addition of 500 mM Tris pH 7.4. Eluted phages were used to infect TG1 cells, superinfection with the M13K07 helper phages allowed re-amplification of the phages for the next round of panning. Enrichment of the library for antigen-specificity was assessed by phage ELISA using a horseradish peroxidase–conjugated anti-M13 antibody (Amersham Biosciences). As soon as enrichment was observed in the phage ELISA, individual colonies of transfected TG1 cells were picked to evaluate the expression of antigen-specific Nbs upon overnight induction with 1 mM isopropyl-β-d-thiogalactopyranoside (IPTG). Periplasmic extracts were prepared using a sucrose-based osmotic treatment and were tested for antigen recognition in ELISA using an anti-HA tag antibody (HA.11 clone 16B12, Covance) and a peroxidase-conjugated anti-mouse IgG (FAb)$_2$ (AbD Serotec) for Nb detection. ELISA development was with ABTS substrate (KPL) and absorbance was measured at 405 nm. To evaluate binding onto live trypanosomes, periplasmic extracts were also used in flow cytometry studies using $2 \times 10^5$ *T. b. brucei* PF parasites for each staining reaction. Parasites were incubated for 30 minutes with 20 µl periplasmic extract in a total volume of 120 µl in round bottom 96-cell plates and were washed with SDM-79 / 10% FCS followed by centrifugation for 10 min at $870 \times g$. Nbs bound to the trypanosome surface were detected using an Alexa Fluor 488 labeled anti-HA tag antibody (1:500 dilution, Covance). Acquisitions were made on a FACSVerse flow cytometer (BD Biosciences) and data analysed using the BD FACSuite software package. Positive clones were subjected to colony PCR (32 cycles 1 min at 94˚C, 1 min at 55˚C and 1 min at 72˚C) using primers MP57 (5'-TTATGCTTCCGGCTCGTATG-3') and GIII (5'-CCACAGACAGCCCTCATAG-3'), followed by ExoSAP-IT treatment (Invitrogen) and sequence analysis of the amplicons using MP57 as sequencing primer (GSF, UA-VIB).

   **Panning against procyclin.**   Four consecutive rounds of panning and phage amplification were performed with an enrichment onto solid phase (Maxisorb immunoplates, Nunc) immobilized procyclin. Overcoating was altered at each round to prevent the selection of overcoat-specific Nbs [milk, bovine serum albumin (BSA), casein, fetal bovine serum]. Individual clones were selected and evaluated in a periplasmic extract ELISA for binding onto procyclin absorbed into ELISA plates. Nb clones that achieved a ratio of > 2 for the optical density on the procyclin coat versus the overcoat were subjected to colony PCR using the MP57 and GIII primers and sequencing of the phagemid insert.

   **Panning against procyclic membrane extract.**   Four consecutive rounds of panning and phage amplification were performed similar as for procyclin using solid phase immobilized

membrane extract 2 as bait. Similar as in the previous panning approach, the overcoating reagent was changed at each round [milk, bovine serum albumin (BSA), casein, fetal bovine serum, BSA]. Positive clones were identified based on a periplasmic extract ELISA using PF membrane extract 2 as antigenic coat.

**Panning against solid phase immobilized procyclic trypanosomes.** For this purpose, *T. b. brucei* PF parasites from cultures were washed twice in PBS and $2.5 \times 10^5$ parasites were allowed to adhere for 30 minutes in the wells of an ELISA plate to serve as bait for the Nb-displaying phages. The first panning round was performed including a negative selection for 30 minutes on skimmed milk that also served as overcoating, the two subsequent panning rounds were performed using milk and BSA as overcoating reagents and allowing phages to bind 30 minutes onto the immobilized trypanosomes. Elution conditions and phage amplification were as those described for the two above procedures. Individual periplasmic extracts were tested in an ELISA using respectively the entire parasites and membrane extract as antigenic coats.

**Panning against live procyclic trypanosomes in suspension.** For the cell panning, $1–2 \times 10^7$ PF trypanosomes were harvested by centrifugation (10 minutes at $870 \times g$), resuspended in 900 μl fresh SDM79 / 10% FCS and chilled on ice. $10^{11}$ phages were added in 1 ml SDM79 / 10% FCS to the PF trypanosomes and incubated by rotation at 4˚C for 1h. Unbound phages were removed by 5 washes with 1.5 ml PBS 10% FCS, each time with a 30" centrifugation at $16,000 \times g$ and at 4˚C. Bound phages were eluted by the addition of 250 μl 1% triethylamine (TEA) solution pH 10. After 7 min, eluted phages were harvested in the supernatant after a 2 min centrifugation at 4˚C. The pH was neutralized by the addition of 250 μl 1 M Tris pH 7.4. Eluted phages were used to infect TG1 cells for re-amplification and the next round of panning. Four panning rounds were performed under the same experimental conditions. Periplasmic extracts of individual colonies were analyzed in a periplasmic extract ELISA against membrane extract 1&2 and in a flow cytometry experiment using live PF *T. b. brucei* parasites from *in vitro* cultures and the Alexa Fluor 488-labeled anti-HA tag antibody for the detection of surface-bound Nbs.

**Panning against live, trypsinized procyclic trypanosomes in suspension.** For cell panning against live, trypsinized PF trypanosomes, $1–2 \times 10^7$ procyclic trypanosomes were harvested by centrifugation (10 minutes at $870 \times g$), and treated for 10 minutes by 0.05% trypsin / EDTA (Gibco) at room temperature to mimic trypsin-mediated trimming of the trypanosomal surface antigens as it is considered to occur in the tsetse fly midgut environment. Subsequently, trypanosomes were washed twice with SDM-79 / 10% FCS medium and the panning procedure was continued exactly as described for the non-trypsinized parasites. Periplasmic extracts of individual colonies were analyzed by flow cytometry for Nb-binding onto live, trypsinized *T. b. brucei* PF parasites.

## Expression and purification of nanobodies

Selected positive clones were expanded in Terrific Broth (TB) medium, followed by the overnight induction with 1 mM IPTG for Nb expression. Nbs were purified by Ni-NTA affinity chromatography (Qiagen), using 0.5 M imidazole in PBS for elution, followed by Nb purification in PBS on a Superdex 75 (10/300) GL column (GE Healthcare) connected to an Akta Purifier 10 apparatus (GE Healthcare). Protein concentrations were determined by the optical density at 280 nm and the individual theoretical extinction coefficients calculated using the ProtParam webtool. Nbs were aliquoted and stored at -20˚C until further use.

## Nanobody binding experiments

For binding studies, purified Nbs were chemically conjugated to Alexa Fluor 488 (Molecular probes), followed by size exclusion of the free label on a Superdex 75 10/300 GL column (GE

Healthcare) connected to an Akta Purifier 10 (GE Healthcare). Alternatively, binding was evaluated using a two-step detection using the Alexa Fluor 488-labeled anti-HA tag antibody that was also used during the Nb selection procedure. Binding was evaluated onto live PF *T. b. brucei*, either or not subjected to a mild trypsin treatment (10 minutes by 0.05% trypsin / EDTA) and onto 2.5% p-formaldehyde fixed *T. b. brucei* PF parasites. Flow cytometry analyses were performed on a FACSVerse flow cytometer (BD Biosciences) and histograms were prepared using FlowJo software (Becton Dickinson, San Jose, CA).

## In vitro nanobody-trypanosome toxicity assays

The anti-trypanosomal activity of the individual purified Nbs was first evaluated *in vitro*. For this purpose, $2 \times 10^6$ *T. b. brucei* PF parasites were suspended in 1 ml SDM-79 medium / 10% FCS and were seeded in 24-well cell culture plates. Selected Nbs were added in different concentrations (10, 12.5, 25 and 50 μg/ml) to individual wells followed by incubation at 27˚C. The parasite densities were monitored at 24 h intervals over a period of 5 days using disposable Uriglass hemocytometers (Menarini Diagnostics). The percentage survival was calculated relative to the condition without Nb (PBS control).

## Recombinant symbiont introduction and fly infections

For the Nb feeding infection experiments, freshly emerged *G. m. morsitans* flies were offered a parasitized blood meal consisting of defibrinated horse blood mixed with $5 \times 10^5$ trypanosomes/ml (intermediate/stumpy forms) and 25 μg/ml or 75 μg/ml of the respective Nbs. Only fully engorged flies were further maintained at 26˚C and 65% relative humidity and were fed two more Nb-supplemented blood meals. Eight days after the infective blood meal, individual flies were analyzed for trypanosome midgut establishment by microscopical evaluation of their midguts.

To evaluate the impact of genetically transformed Nb-expressing *S. glossinidius* strains on fly susceptibility to trypanosome infection, third-instar larvae, collected immediately after larviposition, were injected with $5 \times 10^6$ CFU WT *S. glossinidius* (control) or rec*Sodalis* expressing Nb_46, Nb_19, Nb_88 using 5 μl Hamilton 75RN microsyringes with gauge 34 removable electrotapered needles. Teneral male flies emerged from these larvae were offered a first blood meal of defibrinated horse blood mixed with $5 \times 10^5$ trypanosomes/ml (intermediate/stumpy bloodstream forms). For this, parasitized blood was harvested with heparin from cyclophosphamide-immune suppressed mice (Endoxan, Baxter) 6 days post-infection and mixed with defibrinated horse blood (E&O Laboratories) to obtain $> 10^6$ bloodstream form (BSF) trypanosomes/ml with 80% intermediate/stumpy forms in the infectious blood meal. Only fully engorged flies were further maintained at 26˚C and 65% relative humidity and were fed on uninfected defibrinated horse blood three times per week using an artificial membrane feeding system until dissection. Only male flies were used in the experimental set-up because the susceptibility of female tsetse flies to infection has been suggested to be influenced by the presence of a developing larva in the female abdomen as a result of the tsetse's viviparous reproduction mode [27].

To evaluate the parasite load and symbiont densities, fly midguts were dissected in ATL buffer followed by DNA extraction. A standard curve, generated from extracted procyclic trypanosomal DNA ranging from $3.75 \times 10^5$ parasites/ml to $1.95 \times 10^2$ parasites/ml allowed to estimate the corresponding number of parasites present in the tsetse midgut. Used primers target the *T. b. brucei* ribosomal *18S* gene: *18S_Fw* (5'-TGGGGACAGTACGATGGCAGAGC-3') and *18S_Rev* (5'-TCATAGGCGGTCGGGGATAATTGCG-3'). A standard curve, generated based on a serial dilution series (1:10) prepared using DNA extracted from rec*Sodalis* cultures

ranging from $10^2$ CFU/ml to $10^7$ CFU/ml, allowed to estimate the corresponding number of recSodalis present in the different tsetse fly tissues. Used primers target the pelB and HA region common to all Nb genes: Nb_PelB_Fw, 5'- ATTGTTATTACTCGCGGCCCA -3' and Nb_HA_Rev, 5'- GGAACCGTAGTCCGGAACG -3'. Real-time qPCR (40 cycles 10" at 95°C, 10" at 60°C and 30" at 72°C; 0.3 µM primer concentration) was carried out on a LightCycler (Roche Diagnostics, Mannheim, Germany) in 96-well plates with samples tested in technical duplicates.

## Supporting information

**S1 Fig. Western blot showing secretion and accumulation of Nbs conjugated to a pelB secretion signal by transformed *S. glossinidius* strains in spent media; Lane 1, MW marker, Lane 2, recSodalis::Nb_An46; Lane 3, recSodalis::Nb_88; Lane 4, recSodalis::Nb_19 and Lane 5, recSodalis::Nb_63.** Concentrated spent medium was tested using an anti-His antibody.
(TIF)

**S2 Fig.** Growth curve analysis of *S. glossinidius* expressing recombinant Nb_An46 (green curve), Nb_88 (orange curve), Nb_19 (black curve), Nb_63 (blue curve) and WT *Sodalis* (red curve). The error bars show the ± SD of two biological replicates. Samples were taken every 24h except during exponential growth (0h-48h), 2 samples/24h were taken.
(TIF)

**S3 Fig. Characteristics of recSodalis::Nb_An46 in teneral male flies.** Number of recSodalis:: Nb_An46 present in abdomen and thorax of teneral male flies emerged from larvae injected with 5 x $10^6$ recSodalis::Nb_An46 versus the total number of *Sodalis* (WT + recSodalis:: Nb_An46). The number of WT and recSodalis::Nb_An46 CFUs was estimated using an already validated quantitative real time-PCR protocol (9). The bars represent the mean total *Sodalis* and recSodalis::Nb_An46 CFUs (+/- SD) present in abdomen and thorax of at least 5 individual flies. The number of CFUs is represented in log scale on the y-axis.
(TIF)

**S4 Fig.** *T. b. brucei* EP-procyclin purification: **(A)** Chromatogram (OD215 nm) illustrating the procyclin purification by hydrophobic interaction chromatography on an octyl sepharose column. Eluted peak fractions were evaluated in an orcinol staining. **(B)** Coomassie and silver stained protein profiles of a soluble procyclic extract, two consecutive membrane extractions and purified EP-procyclin.
(TIF)

**S5 Fig. ELISA-based detection of antibody responses in an alpaca immunized with a mixture of purified EP-procyclin and purified procyclic membrane extract 2.** The graphs depict the reactivity of the conventional (IgG1) and heavy-chain antibody isotypes (IgG2) purified from the serum of immunized (red) and control immunized animals (blue) against (A) soluble procyclic extract, (B) procyclic membrane extract 1, (C) procyclic membrane extract 2 and (D) purified procyclin. Serial ½ serum dilutions were applied to each antigen followed by IgG detection, using an in-house rabbit anti-camel polyclonal IgG and a peroxidase-conjugated anti-rabbit IgG (Sigma).
(TIF)

**S6 Fig.** Panning of the anti-procyclic *T. b. brucei* surface Nb library against purified EP-procyclin: (A) Overview of the different panning rounds (Rx) with numbers of positive clones and the resulting clone and family diversity. (B) Enrichment of procyclin-specific phages

throughout the different panning rounds determined by phage ELISA. (C) Flow cytometry analysis to evaluate binding activity of the individual purified Nbs (n = 7) selected onto live parasites, revealed by a one-step detection using Nb-Alexa Fluor 488 conjugates (expressed as percent positive relative to non-stained control population). (D) Alignment of the different selected Nb clones with a corresponding maximum likelihood tree.
(TIF)

**S7 Fig.** Panning of the anti-procyclic surface *T. b. brucei* Nb library against the second procyclic membrane extract: (**A**) Overview of the different panning rounds with numbers of positive clones and the resulting clone and family diversity. (**B**) Enrichment of specific phages reactive against the procyclic membrane components throughout the different panning rounds determined by phage ELISA. (**C**) Flow cytometry analysis to evaluate binding activity of the individual purified Nbs (n = 18) onto live parasites, revealed by a one-step detection using Nb-Alexa Fluor 488 conjugates (expressed as percent positive relative to non-stained control population). (**D**) Alignment of the different selected Nb clones with a corresponding maximum likelihood tree.
(TIF)

**S8 Fig.** Panning of the anti-procyclic surface *T. b. brucei* Nb library against live procyclic trypanosomes in suspension (**A**) Overview of the different panning rounds with numbers of positive clones and the resulting clone and family diversity. (**B**) Enrichment of specific phages reactive against the different procyclic membrane extracts and EP-procyclin (EP) throughout the different panning rounds determined by phage ELISA. (**C**) Flow cytometry analysis to evaluate binding activity of the individual periplasmic extracts onto live procyclic trypanosomes, either (left panel) or not (right panel) pre-treated with trypsin to mimic surface antigen trimming in the tsetse midgut (expressed as percent positive relative to non-stained control population). Nbs bound to the trypanosome surface were detected using an Alexa Fluor 488 labeled anti-HA Tag antibody (1/500 dilution, Covance). (**D**) Alignment of the different selected Nb clones with a corresponding maximum likelihood tree
(TIF)

**S9 Fig.** Panning of the anti-procyclic surface *T. b. brucei* Nb library against live, trypsinized procyclic trypanosomes in suspension (**A**) Overview of the different panning rounds with numbers of positive clones and the resulting clone and family diversity. (**B**) Enrichment of specific phages reactive against the different procyclic membrane extracts and procyclin throughout the different panning rounds determined by phage ELISA. (**C**) Flow cytometry analysis to evaluate binding activity of the individual periplasmic extracts onto live, trypsinated procyclic trypanosomes (expressed as percent positive relative to non-stained control population). Nbs bound to the trypanosome surface were detected using an Alexa Fluor 488 labeled anti-HA Tag antibody (1/500 dilution, Covance). (**D**) Alignment of the different selected Nb clones with a corresponding maximum likelihood tree.
(TIF)

**S10 Fig. ALEXA-labelled Nb_19 was incubated with cultured PF trypanosomes.** Analysis of the samples by confocal microscopy showed a staining of the flagellum. DAPI (blue) stained the nucleus and kinetoplast.
(TIF)

**S11 Fig. PCR analysis of recombinant *S. glossinidius* clones using primers that flank the Tn7 insertion point (i.e. 25 nucleotides downstream of the *glmS* gene.** The recombinant clones all have an approximately 1265 bp higher molecular-weight band compared to the WT

clone resulting from the insertion of the lac promoter:Nb cassette (approximately 825 bp) + transposon ends Tn7L and Tn7R (440 bp).
(TIF)

## Author Contributions

**Conceptualization:** Linda De Vooght, Guy Caljon, Jan Van Den Abbeele.

**Data curation:** Linda De Vooght.

**Formal analysis:** Linda De Vooght, Guy Caljon, Jan Van Den Abbeele.

**Funding acquisition:** Jan Van Den Abbeele.

**Investigation:** Linda De Vooght, Benoît Stijlemans, Guy Caljon, Jan Van Den Abbeele.

**Methodology:** Linda De Vooght, Karin De Ridder, Shahid Hussain, Benoît Stijlemans, Guy Caljon, Jan Van Den Abbeele.

**Project administration:** Jan Van Den Abbeele.

**Resources:** Patrick De Baetselier, Jan Van Den Abbeele.

**Software:** Linda De Vooght, Guy Caljon.

**Supervision:** Guy Caljon, Jan Van Den Abbeele.

**Validation:** Linda De Vooght, Guy Caljon, Jan Van Den Abbeele.

**Visualization:** Linda De Vooght, Guy Caljon.

**Writing – original draft:** Linda De Vooght, Guy Caljon, Jan Van Den Abbeele.

**Writing – review & editing:** Linda De Vooght, Benoît Stijlemans, Patrick De Baetselier, Guy Caljon, Jan Van Den Abbeele.

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
