## [Decision Letter · Decision Letter 0]

13 Dec 2021

Dear Dr. De Vooght,

Thank you very much for submitting your manuscript "Targeting the tsetse-trypanosome interplay using genetically engineered Sodalis glossinidius" for consideration at PLOS Pathogens. As with all papers reviewed by the journal, your manuscript was reviewed by members of the editorial board and by several independent reviewers. The reviewers appreciated the attention to an important topic. Based on the reviews, we are likely to accept this manuscript for publication, providing that you modify the manuscript according to the review recommendations.

Sincerely,

Elizabeth A. McGraw, PhD

Associate Editor

PLOS Pathogens

David Sacks

Section Editor

PLOS Pathogens

Kasturi Haldar

Editor-in-Chief

PLOS Pathogens

orcid.org/0000-0001-5065-158X

Michael Malim

Editor-in-Chief

PLOS Pathogens

orcid.org/0000-0002-7699-2064

Reviewer Comments (if any, and for reference):

Reviewer's Responses to Questions

**Part I - Summary**

Reviewer #1: It is an excellent, well-developed scientific work with well-defined objectives and an experimental system coordinated and consequent to them.

I have no particular remarks on the more strictly technical aspects although I would have liked some more details, at least in the discussion phase, on the reason for a rather limited number of samples analyzed and above all on the possible involvement of the genetic background of the insect (different strains of the vector could they offer different results?).

The main points are on the drafting of the manuscript: in the introduction many of the results are anticipated, consequently a substantial revision of the introduction could offer a better fluidity of reading and consequently an easier understanding of the technical aspects.

Similarly, many technical details are included in the results that are redundant with respect to the correct descriptions placed in the M&M section.

Reviewer #2: This manuscript describes an attempt to develop a new approach to interfere with Trypanosoma development in the tsetse fly, by engineering a fly symbiont to produce antibodies that recognize parasite surface molecules.

I have a few concerns, mostly related to format.

• The experiments are heavily biased to find nanobodies (Nbs) to procyclin, without providing the reader a rationale. It is only much later, in the Discussion, that this is addressed. Given that the majority of the readers are likely to not be Trypanosoma experts, this rationale needs to be explained up front.

• Line 194. “The second membrane extract was found to be highly enriched in membrane proteins with a lower abundance of soluble components as compared to the first membrane extraction (Suppl. Fig. 4B)”. I could not understand how this conclusion was made based on the data shown in the figure.

• Lines 197–200. I could not understand Suppl. Fig. 5.

• I could not understand Suppl. Figs. 6C, 7C, 8C and 9C.

• Suppl. Figs. 7 and 8. The legends say “Panning of the anti-procyclic T. b. brucei Nb library against the second procyclic membrane extract” (Fig. 7) or “….against the live procyclic trypansomes in suspension” (Fig. 8). This lack of clarity is confusing. It does not make sense to screen a library made against a single protein (procyclin) against a complex collection of antigens (membrane extract or live parasites). It is important to be precise and give the library a name that reflects its true origins (it was made by immunizing the alpaca not with procyclin alone, but with a mixture of this protein and a membrane extract).

• Suppl. Figs. 7B and 8B. Define “EP”.

• Line 244. Explain the rationale for trypsinization.

• Line 253. Explain the criteria used for the selection of clones for protein production in E. coli.

• Figure 2. Expand the legend to explain how the measurements were made. While I could understand what the red lines represent, I could not figure out neither from the legend nor from the Materials & Methods how the blue lines were generated. How were control cells stained?

• Line 329. Reference 13 is not appropriate. This article describes about the POSSIBILITY of using viruses for paratransgenesis, but never showed that it works. I am not aware of any subsequent publications that use this virus vector for paratransgenesis.

• Fig. 3. State the number of independent experiments that the data in panels A and B represent. For panel A, consider signaling in the figure which data are from the first experiment and which are from the second experiment.

• Figs. 4 AND 5. 1) Indicate the number of independent experiments that the data represent; 2)indicate the number of midguts assayed; 3) indicate what the horizontal bars represent (means?, medians?, standard deviations?).

• Figure 5. Please state in the legend or in the text, the median (or mean) percent transgenic bacteria for both Nb_88 and Nb_19.

• For the benefit of the non-specialist, consider explaining why only male flies were used.

Overall, the experiments seem to be well executed and the findings should be of interest to the journal readers.

Reviewer #3: This paper describes the use of paratrangenesis approach to produce tsetse flies refractory to Trypanosoma brucei brucei procyclic developmental stage through the production of effector molecules that can target the trypanosome-tsetse fly crosstalk via the genetically modified Sodalis. The results indicate that although genetically engineered S. glossinidius expressing Nb_88 significantly compromised parasite development in the tsetse fly midgut both at the level of infection rate and parasite load, the expression of Nb_19 by S. glossinidius resulted in a significantly enhanced midgut establishment. The results are of interest as it provides the proof-of-principle for the use of paratrangenesis to modify the susceptibility of tsetse flies to Trypanosoma infection and the represent a major step forward in the development of a control strategy based on paratransgenic tsetse flies. The manuscript well written. I do have some minor comments, which I fell need to be addressed before considering this article for publication.

**Part II – Major Issues: Key Experiments Required for Acceptance**

Reviewer #1: (No Response)

Reviewer #2: (No Response)

Reviewer #3: No major issues found

**Part III – Minor Issues: Editorial and Data Presentation Modifications**

Reviewer #1: As stated above, I suggest to revise mainly the introduction in which many of the results are anticipated. A substantial revision of the introduction could offer a better fluidity of reading and consequently an easier understanding of the technical aspects.

Similarly, I suggest also a revision of the result section where too many technical details are included being redundant with respect to the correct descriptions placed in the M&M section.

Reviewer #2: (No Response)

Reviewer #3: Minor comments

1- As general comments, the materials and method need revision to indicate in many sections, details on the source of the materials used i.e. kits plasmids and reagent, volume of reaction etc., to ensure the reputability of the work.

2- It seems there a common typo that the time in min was indicated as (’) i.e. in lines 510, 513, 515, 540, 559, 588, 607, 609,616, 617, 618, 646, please correct.

3- In line 537: how many washes were used, please indicate the number.

4- Line 545: indicate the volume or the quantity of the DNA template.

5- Line 548: indicate the source of the pMECS and pHEN4.

6- Line 551-552: It is not clear how the digestion with Xho1 will reduce the self-ligation, please clarify.

7- Line 553-554: please indicate the source of the T4 Ligase and the final volume of the reaction.

8- Line 555: indicate the volume of the elution buffer used after the wash with QIAquick. Indicate the devise use for the electroporation.

9- Line 556: do you mean competent cells?

10- Line 556: was the incubation under shaking?

11- Line 676-682: was the qPCR data normalized against a house keeping gene to eliminate the pitting error?

12- The resolution of figure 1B, 3A, need improvement.

13- The scientific name in the following references need to be italic (1, 3, 7, 8, 10, 11, 13, 17, 19, 20, and 23)

PLOS authors have the option to publish the peer review history of their article (what does this mean?). If published, this will include your full peer review and any attached files.

Reviewer #1: No

Reviewer #2: No

Reviewer #3: No

Figure Files:

Data Requirements:

Reproducibility:

References:

---

## [Editor Report · Decision Letter 1]

15 Feb 2022

Dear Dr. De Vooght,

We are pleased to inform you that your manuscript 'Targeting the tsetse-trypanosome interplay using genetically engineered Sodalis glossinidius' has been provisionally accepted for publication in PLOS Pathogens.

Best regards,

Elizabeth A. McGraw, PhD

Associate Editor

PLOS Pathogens

David Sacks

Section Editor

PLOS Pathogens

Kasturi Haldar

Editor-in-Chief

PLOS Pathogens

orcid.org/0000-0001-5065-158X

Michael Malim

Editor-in-Chief

PLOS Pathogens

orcid.org/0000-0002-7699-2064
---

## [Editor Report · Acceptance letter]

2 Mar 2022

Dear Dr. De Vooght,

We are delighted to inform you that your manuscript, "Targeting the tsetse-trypanosome interplay using genetically engineered Sodalis glossinidius," has been formally accepted for publication in PLOS Pathogens.

Best regards,

Kasturi Haldar

Editor-in-Chief

PLOS Pathogens

orcid.org/0000-0001-5065-158X

Michael Malim

Editor-in-Chief

PLOS Pathogens

orcid.org/0000-0002-7699-2064